# Apelin-13 Inhibits Methylglyoxal-Induced Unfolded Protein Responses and Endothelial Dysfunction via Regulating AMPK Pathway

**DOI:** 10.3390/ijms21114069

**Published:** 2020-06-06

**Authors:** Sujin Kim, Suji Kim, Ae-Rang Hwang, Hyoung Chul Choi, Ji-Yun Lee, Chang-Hoon Woo

**Affiliations:** 1Department of Pharmacology and Smart-Aging Convergence Research Center, Yeungnam University College of Medicine, 170 Hyeonchung-ro, Nam-gu, Daegu 42415, Korea; sujin8810@naver.com (S.K.); rlatnwl@yu.ac.kr (S.K.); dofkddofkd@hanmail.net (A.-R.H.); hcchoi@med.yu.ac.kr (H.C.C.); 2College of Pharmacy, Chung-Ang University, 84 Heukseok-ro, Dongjak-gu, Seoul 06974, Korea

**Keywords:** methylglyoxal, apelin, unfolded protein response, endothelial dysfunction, AMPK

## Abstract

It has been suggested that methylglyoxal (MGO), a glycolytic metabolite, has more detrimental effects on endothelial dysfunction than glucose itself. Recent reports showed that high glucose and MGO induced endoplasmic reticulum (ER) stress and myocyte apoptosis in ischemic heart disease was inhibited by apelin. The goal of the study is to investigate the molecular mechanism by which MGO induces endothelial dysfunction via the regulation of ER stress in endothelial cells, and to examine whether apelin-13, a cytoprotective polypeptide ligand, protects MGO-induced aortic endothelial dysfunction. MGO-induced ER stress and apoptosis were determined by immunoblotting and MTT assay in HUVECs. Aortic endothelial dysfunction was addressed by en face immunostaining and acetylcholine-induced vasodilation analysis with aortic rings from mice treated with MGO in the presence or absence of apelin ex vivo. TUDCA, an inhibitor of ER stress, inhibited MGO-induced apoptosis and reduction of cell viability, suggesting that MGO signaling to endothelial apoptosis is mediated via ER stress, which leads to activation of unfolded protein responses (UPR). In addition, MGO-induced UPR and aortic endothelial dysfunction were significantly diminished by apelin-13. Finally, this study showed that apelin-13 protects MGO-induced UPR and endothelial apoptosis through the AMPK pathway. Apelin-13 reduces MGO-induced UPR and endothelial dysfunction via regulating the AMPK activating pathway, suggesting the therapeutic potential of apelin-13 in diabetic cardiovascular complications.

## 1. Introduction

Diabetes mellitus is considered as an independent risk factor for cardiovascular complications in epidemiological studies [1]. It has been revealed that a highly elevated glucose level caused by diabetes mellitus plays a major role in diabetic cardiovascular complications. Nevertheless, the role of other glucose metabolites in diabetes mellitus and its complications is still uncertain. During glycolysis with other metabolic pathways, glucose can form several dicarbonyl metabolites such as glyoxal, methylglyoxal (MGO), and 3-deoxyglucosone [2]. These dicarbonyl metabolites react with proteins or lipids to form advanced glycation end products (AGEs) [3,4]. Among them, under diabetic conditions, the formation of MGO which is highly reactive is increased [5,6]. Even though MGO is the precursor of AGEs, recent studies demonstrated that MGO had a greater potential to stimulate vascular damage than AGEs and glucose itself, indicating that MGO is involved in the development of diabetic cardiovascular complications [7,8]. However, the underlying mechanisms of MGO-induced endothelial dysfunction have not been identified yet.

Endoplasmic reticulum (ER) stress has been known as a critical event involved in triggering diabetes and diabetic complications [9,10]. Newly synthesized protein must be folded into functional configurations in the ER that are regulated by a complex cast of machinery [11]. This machinery senses the status of protein folding and then responds to the ER transmembrane sensing receptors by adjusting the capacity of the system, thus the protein folding demand can keep a homeostatic balanced status [12]. These ER sensing receptors to unfolded proteins initiate the ER stress response to restore normal ER function. Three major sensing proteins initiate unfolded protein responses (UPR) through the activation of the transcription factors leading to induction of genes encoding ER-targeted chaperones, calcium-binding proteins, and disulfide isomerases. In contrast to the gene induction system, through PERK-eIF2α cascade, ER stress suppresses and regulates overloaded protein synthesis on the ER protein folding machinery. However, if the unmanageable demand of protein synthesis is imposed on the ER folding machinery, the cell follows the degenerative pathway leading to apoptosis [11]. There are three major pathways related to ER stress which are protein kinase dsRNA-activated protein kinase-like ER kinase (PERK), inositol-requiring protein 1α (IRE1α), and the transcription factor activating transcription factor 6 (ATF6) [13]. Among them, the PERK pathway is involved in apoptosis via ATF4 activation and subsequent CHOP induction [14]. Recently, it has been reported that MGO induces ER stress and leads cells to apoptosis in cardiomyocytes [15]. Nevertheless, the specific molecular mechanism by which MGO controls endothelial dysfunction and ER stress is still unknown. Recent studies have shown that MGO increases in intracellular levels of reactive oxygen species (ROS) [16]. In addition, it has been suggested that ROS induced ER stress responses [17]. Interestingly, our previous study showed that activated protein C attenuated MGO-induced ROS generation in cardiomyocytes [18]. Thus, it is reasonable to make a hypothesis that ROS might be involved in the regulation of MGO-induced ER stress and endothelial apoptosis.

Apelin is the endogenous polypeptide ligand for the previously orphaned G protein-coupled receptor, APJ [19]. It has been emerging that apelin is a key regulator of cardiovascular homeostasis [20,21,22]. Animal studies showed that apelin increased cardiac contractility and reduced ventricular preload and afterload in rats with failing hearts. Interestingly, Tao et al. reported that apelin-13 protected the heart against ischemia-reperfusion injury via inhibition of ER stress [23]. Additionally, apelin-13 treatment ameliorated diabetes-induced reduction in islet mass and insulin content in Akita mouse which is a mouse model for ER stress-induced diabetes [24]. These reports suggested that ER stress is a key regulator in apelin-mediated protective effects in diabetes and heart failure. In addition to the protective role of apelin in heart failure and diabetes, it has been reported that apelin induced nitric oxide-dependent vasodilation in humans [25,26,27]. Moreover, Yang and colleagues suggested that apelin improves angiotensin II-induced endothelial cell senescence [28]. Therefore, it was hypothesized that apelin might protect MGO-induced endothelial dysfunction via downregulating ER stress.

The aims of the present study were to investigate the mechanism of MGO-induced endothelial dysfunction, and the cytoprotective role of apelin-13 in MGO-induced UPR and aortic endothelial dysfunction.

## 2. Results

### 2.1. MGO Induces Endothelial Apoptosis in an ER Stress-Dependent Manner in HUVECs

To identify whether ER stress is induced by MGO, four different signaling pathways related UPR were detected by immunoblotting against each marker protein in human umbilical vein endothelial cells (HUVECs). Protein expressions of PERK-eIF2α-ATF4-CHOP pathway and JNK pathway were increased by 100 µM MGO in a time-dependent manner (Figure 1A,B). However, signaling pathways of XBP-1 and ATF6 were not affected in response to MGO (Figure 1C,D). In addition, we found that 100 µM is the most effective dose of MGO (Figure 1E).

To determine the role of UPR when HUVECs are exposed to MGO, HUVECs were pretreated by TUDCA which is a chemical inhibitor of ER stress. As shown in Figure 1F, induced cleaved forms of PARP-1 and caspase-3 by MGO were inhibited by TUDCA pretreatment. MGO reduced cell viability and it was significantly recovered by TUDCA, consistent with the Western blotting data (Figure 1G). These results indicate that MGO induces endothelial apoptosis via regulation of UPR.

### 2.2. Apelin-13 Ameliorates MGO-Induced UPR and Apoptosis in HUVECs and Aortic Endothelial Dysfunction Ex Vivo

The involvement of UPR in the protective effects of apelin-13 against MGO was examined by immunoblotting assay. As shown in Figure 2A, protein expressions of ATF4 and CHOP induced by MGO were inhibited by apelin-13 in a dose-dependent manner. To examine the cytoprotective role of apelin-13, it was investigated whether apelin-13 could protect endothelial cells from apoptosis induced by MGO. In Figure 2B, apoptotic markers induced by MGO were markedly inhibited by apelin-13. In addition, the cytoprotective effect of apelin-13 in MGO-induced reduction of cell viability was confirmed by MTT assay (Figure 2C), suggesting the involvement of UPR regulation by apelin-13 in the MGO-induced endothelial apoptosis.

To determine the role of apelin-13 on MGO-induced endothelial apoptosis ex vivo, aorta from C57BL/6 mice were treated with MGO (100 µM) for 24 h in the presence or absence of 1 µM apelin-13. Aortic endothelial cells were stained with anti-VE-cadherin antibody for endothelial cell junction and TUNEL for apoptosis. MGO-induced aortic endothelial apoptosis was markedly inhibited by apelin-13 (Figure 3A). In addition, the effect of MGO in acetylcholine-induced vasorelaxation was determined by aortic ring assay. As shown in Figure 3B, MGO-induced endothelial dysfunction was significantly improved by apelin-13. However, sodium nitroprusside (nitric oxide donor)-mediated vasorelaxations were not affected by MGO in the presence or absence of apelin-13 (Figure 3C). These results suggest that apelin-13 protects aortic endothelial apoptosis and dysfunction against MGO ex vivo.

### 2.3. MGO-Induced URP and Apoptosis Was Regulated via ROS

To examine the involvement of reactive oxygen species (ROS) in MGO-induced UPR and apoptosis, HUVECs were pretreated with N-acetyl cysteine (NAC), a ROS scavenger, upon stimulation with MGO. As shown in Figure 4A, NAC reduced MGO-induced apoptotic marker protein expression. Consistent with the Western blotting data, MGO-mediated reduction of cell viability was significantly recovered by NAC (Figure 4B). In addition, MGO-induced ATF4 and CHOP expression was diminished by NAC (Figure 4C). However, apelin-13 did not affect the ROS production induced by MGO suggesting that apelin-13 regulates the downstream event of MGO-induced ROS production (Figure 4D).

### 2.4. Apelin-13 Ameliorates MGO-Induced Endothelial Apoptosis through AMPK Pathway

To identify the molecular mechanism by which apelin-13 regulates the inhibition of MGO-induced apoptosis, HUVECs were pretreated with wortmannin (PI3K inhibitor), U73122 (PLC inhibitor), and compound C (AMPK inhibitor) upon stimulation with MGO in the presence of apelin-13. As shown in Figure 5A, apelin-mediated inhibition of MGO-induced apoptosis was impaired by compound C, but not by wortmannin and U73122. Consistent with the Western blotting data, apelin-mediated cytoprotective effect was significantly impaired by compound C (Figure 5B). In addition, AMPK activation with adenoviral expression of catalytically active form of AMPK inhibited MGO-induced endothelial apoptosis (Figure 5C). Consistent with the Western blotting data, AMPK activation rescued the cell viability under MGO treatment (Figure 5D). These results indicate that apelin-13 has a cytoprotective effect against MGO-induced endothelial apoptosis via regulation of AMPK activating pathway.

## 3. Discussion

In the present study, we investigated the involvement of apelin-13 in MGO-induced UPR and aortic endothelial dysfunction. The major findings of the present study are that apelin-13 inhibits MGO-induced UPR and apoptosis in HUVECs via ER stress (Figure 1 and Figure 2) and ameliorates MGO-induced aortic endothelial dysfunction ex vivo (Figure 3). We also found that apelin-13 inhibits MGO-induced endothelial apoptosis through AMPK pathway (Figure 5). Taken together, we suggest the proposed model of cytoprotective roles of apelin-13 in ER stress and endothelial dysfunction against MGO (Figure 6).

Recent reports showed that apelin has a cardioprotective effect in diabetic cardiovascular complications via inhibiting ER stress [23,24]. As shown in Figure 2, protein expressions of ATF4 and CHOP induced by MGO were inhibited by apelin-13, suggesting that apelin-13 ameliorates MGO-induced apoptosis via downregulating CHOP induction. Given the fact that apelin-13 protected endothelial cells from MGO-induced CHOP expression and apoptosis, apelin-13 might regulate MGO-induced ER stress in endothelial cells. It has been known that apelin activated several signaling pathways including AMPK, PLC, and PI3K-AKT [29,30]. However, the relevance of these signaling pathways in the regulation of ER stress remains unknown. Recent reports have shown that AMPK and AKT pathways regulate ER stress. In addition, AMPK activation has been shown to prevent ER stress-mediated apoptosis [18,31]. In the present study, apelin-mediated inhibition of MGO-induced endothelial apoptosis was impaired in the presence of AMPK inhibitor, but not PLC inhibitor and PI3K inhibitor (Figure 5). These results suggest that apelin-13 ameliorates MGO-mediated CHOP induction and endothelial apoptosis via regulation of AMPK pathway.

Palsamy et al. reported that MGO induced intracellular Ca^2+^ mobilization from ER store which triggers ER stress in lens epithelial cells [32]. However, the role of Ca^2+^ mobilization in MGO-induced ER stress and apoptosis has not been addressed in endothelial cells. It has been established that the mobilization of Ca^2+^ from the ER activates caspase-12/14 and mitochondria-dependent apoptosis. Interestingly, Toltl and colleagues showed that activated protein C reduced thapsigargin-induced UPR and caspase-3 activity in blood monocytes [33]. Since thapsigargin is a chemical ER stress inducer via ER Ca^2+^ flux, it is possible that apelin might regulate MGO-induced ER stress and apoptosis via reducing intracellular Ca^2+^ mobilization in endothelial cells. The molecular mechanism of Ca^2+^ mobilization in the cytoprotective effects of apelin-13 against MGO-induced ER stress and apoptosis remains to be further determined. In addition to intracellular Ca^2+^ mobilization, ROS are involved in MGO-induced UPR and apoptosis. Although NAC, a ROS scavenger, inhibits MGO-induced UPR and apoptosis, apelin-13 did not affect the MGO-induced intracellular ROS generation (Figure 4). These results suggest that apelin-13 inhibits MGO-induced UPR and apoptosis at the downstream level of ROS.

## 4. Materials and Methods

### 4.1. Reagents and Antibodies

Methylglyoxal (MGO), TUDCA, wortmannin, U73122, compound C, and SP600125 were purchased from Sigma (St. Louis, MO, USA). 3-(4,5-Dimethylthiazol-2-yl)-2,5-Diphenyltetrazolium (MTT) reagents and apelin-13 were obtained from Amresco (Solon, OH, USA) and American Peptide (Sunnyvale, CA, USA), respectively. Antibodies were purchased from the following vendors: KDEL (GRP94, GRP78) (Enzo Life Sciences, Lörrach, Germany); ATF4, GADD153 (CHOP), PERK, phospho-PERK, eIF2α, and α-actin (Santa Cruz, CA, USA); PARP-1, cleaved caspase-3, VE-cadherin, phospho-elF2α, JNK, and phospho-JNK (Cell Signaling Technology, Danvers, MA, USA); and α-tubulin (Sigma, St. Louis, MO, USA).

### 4.2. Cell Culture

Human umbilical vein endothelial cells (HUVECs; Lonza, Basel, Switzerland) from passage 5 to 8 were cultured in medium M200 (GIBCO, Gaithersburg, MD, USA) containing 5% fetal bovine serum (FBS) and endothelial growth factor supplement (LSGS; Cascade Biologics, Portland, OR, USA) on 0.2% gelatin coated cell culture dishes. Cells were incubated in a humidified atmosphere containing 5% CO_2_ at 37 °C. Cells were grown until 80% confluence. All compounds were freshly prepared in culture medium without any supplements, sterilized by filtration and added to the cell cultures.

### 4.3. Protein Extraction and Western Blot Analysis

Cells were lysed in radioimmunoprecipitation assay (RIPA) lysis buffer (pH 7.4) supplemented with 50 mM Tris, 150 mM NaCl, 1 mM EDTA, 1 mM PMSF, and 0.01 mM protease inhibitor cocktail. Lysed cells were incubated with slight agitation on ice for 15 min. The insoluble material was removed by centrifugation at 15,000× *g* for 15 min at 4 ℃. Concentrations of cellular protein were determined by Bradford protein assay using bovine serum albumin as a protein standard. Proteins were separated by SDS-PAGE and transferred to polyvinylidene difluoride membrane. Membranes were immunoblotted with primary antibodies (1:1000) indicated in the figures followed by immunoblotting with corresponding secondary antibodies (1:5000). Signals were visualized by using chemiluminescence detection regents (Millipore, Temecula, CA, USA) according to the manufacturer’s instructions.

### 4.4. MTT Assay

HUVECs were cultured on 24-well plates. When the cells are approximately 80% confluent, the media were replaced with M200 supplemented with 5% FBS without endothelial growth factor supplement. After overnight incubation, cells were pretreated with various inhibitors indicated in figures for 1 h and then stimulated with MGO for 24 h. MTT reagents were incubated for 4 h at 37 °C, and then washed with PBS. The precipitates formed were dissolved in DMSO. Cell viability was determined using microplate reader (Bio-Rad, Hercules, CA, USA) at 570 nm.

### 4.5. En Face Experiment for Aorta

To determine the role of apelin-13 on MGO-induced endothelial apoptosis ex vivo, eight-week old male C57BL/6 mice were anesthetized and aorta was isolated. Isolated aorta was cultured in DMEM (GIBCO, Gaithersburg, MD, USA) supplemented with 10% FBS, 50 units/mL penicillin, and 50 µg/mL streptomycin. Aorta treated with MGO (100 µM) for 24 h, with or without apelin-13 (1 µM) pretreatment, was fixed with 4% paraformaldehyde for 5 min and permeabilized with PBS with 0.1% Tween. Following that, the fat was removed and 5% goat serum was used for blocking. Aortic endothelial cells were stained with anti-vascular endothelial-cadherin antibody (1:500) for endothelial cell junction overnight at 4 °C. After washing with PBST, cells were incubated with fluorescein isothiocyanate-conjugated anti-rat IgG (Invitrogen Carlsbad, CA, USA) for 90 min. Apoptosis was measured by TUNEL (terminal deoxyribonucleotide transferase (TdT)-mediated dUTP nick end labeling) staining, which targets in situ DNA fragmentation, using the In Situ Cell Death Detection Kit (Roche, Indianapolis, IN, USA) as described previously [15]. Signals were visualized at ×400 under the confocal microscope (Leica, Bannockburn, IL, USA). Quantification of apoptosis is shown as the percentage of TUNEL positive cells. More than 200 cells were counted for each category. All animal experiments were conducted in accordance with a protocol approved beforehand by the Institutional Animal Care and Use Committee of Yeungnam University College of Medicine, Daegu, Republic of Korea (YUMC-AEC2015-033, 15 February 2016). In addition to this, all experiments were performed in accordance with the relevant guidelines and regulations.

### 4.6. Vascular Reactivity Study (Vessel Tension Response of Aortic Rings)

To determine endothelium-dependent vascular relaxation, thoracic aortas were removed from C57BL/6 mice (8 weeks old) after sacrifice. Adventitial fat and connective tissues were carefully removed and arteries cut into 2 mm rings under a microscope. Furthermore, thoracic aortas were treated with MGO (100 µM) for 24 h, with or without apelin-13 (10 µM) pretreatment. After 24 h, these rings were then suspended by a small vessel wire myograph containing 37 °C Krebs-bicarbonate buffer (117 mM NaCl, 4.8 mM KCl, 1.2 mM MgSO_4_, 25 mM NaHCO_3_, 1.2 mM KH_2_PO_4_, 5.7 mM glucose, 2.5 mM CaCl_2_) in a 95% O_2_/5% CO_2_ atmosphere. Isometric tension was measured using a force transducer. Aortic rings were equilibrated in buffer for 30 min under resting tension of 0.5 g and then constricted by adding phenylephrine (10^−7^ M) until a steady-state was reached. Endothelium-dependent relaxations were assessed by measuring the dilatory response of arteries to acetylcholine (from 10^−9^ M to 10^−6^ M) or sodium nitroprusside (from 10^−9^ M to 10^−8^ M).

### 4.7. Measurement of Intracellular Reactive Oxygen Species (ROS)

ROS arbitrary units were measured using a 2′7′-dichlorofluorescein diacetate (DCF-DA) dye, as described previously [34]. Briefly, HUVECs (2 × 10^5^ cells per well) were seeded onto 6-well plates and cultivated for 24 h. The stimulated cells were washed with PBS and incubated for 10 min with 10 µM DCF-DA dissolved in DMSO. The cells were then washed 3 times with PBS and cells detached by adding trypsin. Then the cells suspended with phenol red free DMEM were subjected to flow cytometry.

### 4.8. Adenoviral Transduction

Adenoviruses expressing the catalytically active forms of the α1 subunit of AMPK (CA-AMPKα1) or LacZ-GFP as a vehicle control were amplified in AD293 cells using standard methodologies [35]. HUVECs were transduced with Ad-LacZ-GFP or Ad-CA-AMPKα1 at a multiplicity of infection (MOI) of 10 for 24 h and then further exposed to 100 µM MGO for 24 h. Transduction efficiency of adenoviruses was determined by visualization of GFP from Ad-LacZ-GFP using the confocal microscope (Leica, Bannockburn, IL, USA). Ten MOI of Ad-LacZ-GFP showed more than 70% of green fluorescence positive cells.

### 4.9. Statistical Analysis

Data in the bar graphs are represented as means ± SDs. Statistical significance of difference was measured by Student’s *t*-test and multiple group comparisons using ANOVA followed by Bonferroni’s post hoc test. A probability value (p values) of <0.05 was considered significant. All results are presented as the means of at least three independent experiments.

## 5. Conclusions

The present study demonstrated the protective effects of apelin-13 in MGO-induced UPR and endothelial dysfunction via the AMPK signaling pathway. In addition, ROS and ER stress might be involved in the regulation of MGO-induced endothelial dysfunction. These results indicate that apelin-13 could be a novel target for therapeutic intervention of endothelial dysfunction related to diabetic vascular complications.

## Figures and Tables

**Figure 1 ijms-21-04069-f001:**
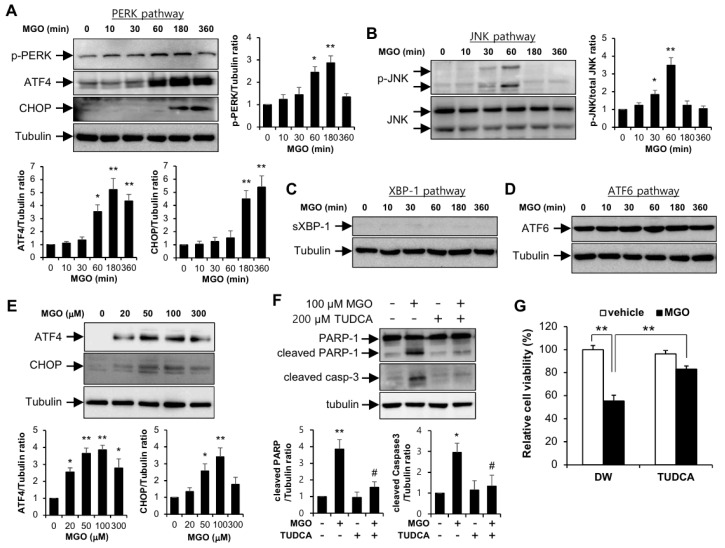
MGO induces endothelial apoptosis via unfolded protein responses in HUVECs. HUVECs were stimulated with 100 µM MGO for the indicated time periods. Protein levels were analyzed by immunoblotting with specific antibodies as follows: (**A**) anti-phospho-PERK, anti-phospho-eIF2α, anti-ATF4, anti-CHOP, and anti-tubulin antibodies; (**B**) anti-phospho JNK, anti-JNK, and anti-tubulin antibodies; (**C**) anti-spliced XBP-1 and anti-tubulin antibodies; (**D**) anti-ATF6 and anti-tubulin antibodies. Bar graphs represent the densitometric results of Western blot bands. * *p* < 0.05 and ** *p* < 0.01 vs. vehicle control (*n* = 3). Results are representative of three independent experiments. (**E**) HUVECs were exposed to the indicated doses of MGO for 6 h and cell lysates were applied for immunoblotting using specific antibodies against ATF-4, CHOP, and tubulin. The graph shows the densitometric quantification of Western blot bands. * *p* < 0.05 and ** *p* < 0.01 vs. vehicle control (*n* = 3). Results are representative of three independent experiments. (**F**) HUVECs were exposed to vehicle or 100 µM MGO in the presence or absence of TUDCA (200 µM). Cells were lysed and then applied for Western blotting analysis. Protein level amounts were determined by immunoblotting with specific antibodies against PARP-1 and caspase-3. The graph shows the densitometric quantification of Western blot bands. * *p* < 0.05 and ** *p* < 0.01 vs. vehicle control, # *p* < 0.05 vs. MGO-treated cells (*n* = 3). Results are representative of three independent experiments. (**G**) MTT assay was performed as described in the Materials and Methods section. Results are expressed as means ± SDs and are representative of three independent experiments. ** *p* < 0.01 (*n* = 3).

**Figure 2 ijms-21-04069-f002:**
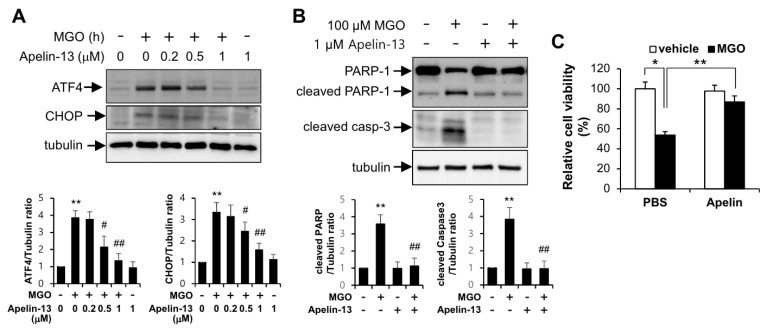
Apelin-13 protects endothelial cells from MGO-induced UPR and apoptosis. HUVECs were pretreated with 1 µM apelin-13 for 1 h followed by exposure to 100 µM MGO for 6 or 24 h. (**A**) Cells were exposed to 100 µM MGO for 6 h in the presence or absence of apelin-13 (0, 0.2, 0.5, 1 µM). Amount of protein expression was determined by immunoblotting with specific antibodies against ATF4, CHOP, and tubulin. Bar graphs represent the densitometric results of Western blot bands. ** *p* < 0.01 vs. vehicle control, # *p* < 0.05; ## *p* < 0.01 vs. MGO-treated cells (*n* = 3). Results are representative of three independent experiments. (**B**) HUVECs were lysed and then applied for Western blotting analysis. Protein level amounts were determined by immunoblotting with anti-PARP-1 and anti-caspase-3 antibodies. The graph shows the densitometric quantification of Western blot bands. ** *p* < 0.01 vs. vehicle control, ## *p* < 0.01 vs. MGO-treated cells (*n* = 3). Results are representative of three independent experiments. (**C**) MTT assay was performed as described in the Materials and Methods section. Results are expressed as means ± SDs and are representative of three independent experiments. * *p* < 0.05; ** *p* < 0.01 (*n* = 3).

**Figure 3 ijms-21-04069-f003:**
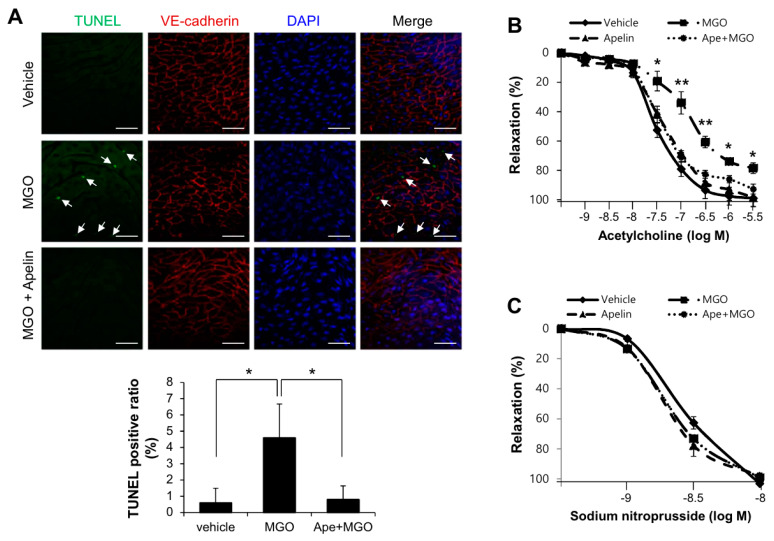
Apelin-13 protects aortic vessels from MGO-induced endothelial dysfunction. To determine the role of apelin-13 on MGO-induced cell apoptosis and endothelial dysfunction ex vivo, C57BL/6 mice were anesthetized and aorta was isolated. (**A**) Aorta treated with MGO (100 µM) for 24 h, with or without apelin-13 (1 µM) pretreatment was stained with anti-VE-cadherin antibody for endothelial cell junction. Apoptosis was measured by TUNEL staining using the In Situ Cell Death Detection Kit (Roche, Indianapolis, IN) as described in the Materials and Methods section. Signals were observed under the confocal microscope (×400). Arrows indicate TUNEL positive cells which are apoptotic endothelial cells. Scale bars: 100 µm. Quantification of apoptosis is shown as the percentage of TUNEL positive cells. More than 200 cells were counted for each category. Data represent means ± SDs. * *p* < 0.05 (*n* = 5). (**B**,**C**) The effect of acetylcholine on phenylephrine-induced vessel contraction in mouse aortic rings. (**B**) Mouse aortic rings were preincubated with phenylephrine (10^−7^ M) and then exposed to acetylcholine (10^−9^–10^−6^ M). The results shown are representative of three independent experiments. * *p* < 0.05; ** *p* < 0.01 vs. MGO treated with apelin-13 group (Ape+MGO). (**C**) Dose-responses to sodium nitroprusside, nitric oxide donor, of phenylephrine-induced precontracted aortic rings. Results are presented as the means ± SDs. *n* = 7 per group.

**Figure 4 ijms-21-04069-f004:**
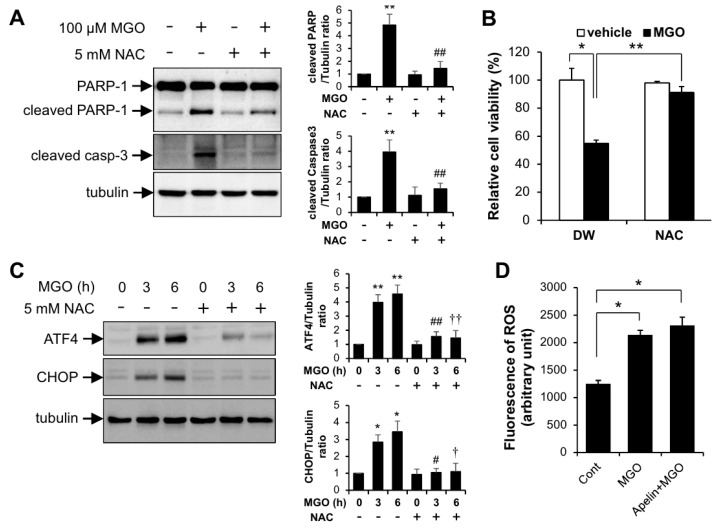
The effect of NAC in MGO-induced UPR and apoptosis in HUVECs. HUVECs pretreated with 5 mM NAC for 1 h were exposed to 100 µM MGO for 24 h. (**A**) HUVECs were lysed and then applied for Western blotting analysis. Protein level amounts were determined by immunoblotting with anti-PARP-1 and anti-caspase-3 antibodies. The graph shows the densitometric quantification of Western blot bands. ** *p* < 0.01 vs. vehicle control, ## *p* < 0.01 vs. MGO-treated cells (*n* = 3). Results are representative of three independent experiments. (**B**) MTT assay was performed as described in the Materials and Methods section. Results are expressed as means ± SDs and are representative of three independent experiments. * *p* < 0.05, ** *p* < 0.01 (*n* = 3). (**C**) Cells were stimulated with 100 µM MGO for the indicated time periods (3 and 6 h) in the presence or absence of 5 mM NAC. Amount of protein expression was determined by immunoblotting with specific antibodies against anti-ATF4, and anti-CHOP and anti-tubulin. Bar graphs represent the densitometric results of Western blot bands. * *p* < 0.05 and ** *p* < 0.01 vs. vehicle control, # *p* < 0.05 and ## *p* < 0.01 vs. MGO-treated cells (3 h), † *p* < 0.05 and †† *p* < 0.01 vs. MGO-treated cells (6 h). *n* = 3 per group. Results are representative of three independent experiments. (D) Cells were stimulated with 100 µM MGO for 6 h in the presence or absence of 1 µM apelin-13. Amounts of ROS were measured by DCF-DA fluorescence staining under flow cytometry as described in the Materials and Methods section. Results are presented as the means ± SDs and are representative of three independent experiments. * *p* < 0.05 vs. vehicle control (*n* = 5).

**Figure 5 ijms-21-04069-f005:**
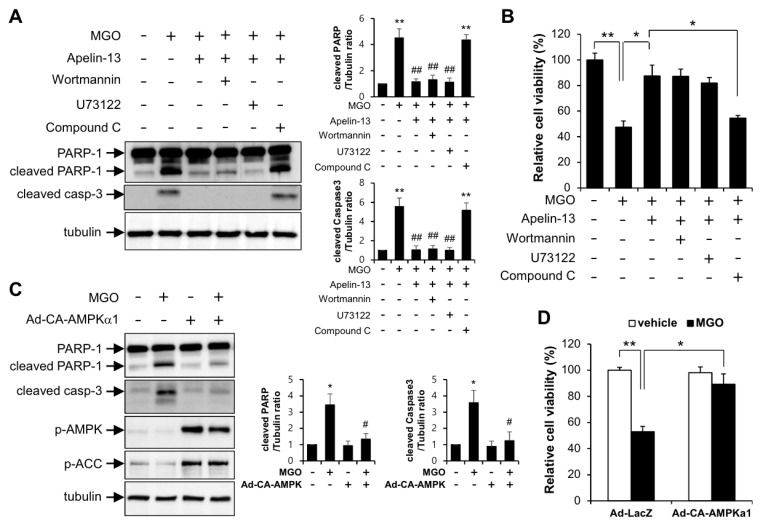
Apelin-13 ameliorates MGO-induced apoptosis through AMPK pathway in HUVECs. Cells pretreated with wortmannin (50 nM), U73122 (1 µM), and compound C (1 µM) were stimulated with 100 µM MGO for 24 h in the presence or absence of 1 µM apelin-13. (**A**) HUVECs were lysed and then applied for Western blotting analysis. Protein level amounts were determined by immunoblotting with anti-PARP-1 and anti-caspase-3 antibodies. The graph shows the densitometric quantification of Western blot bands. ** *p* < 0.01 vs. vehicle control, ## *p* < 0.01 vs. MGO-treated cells (*n* = 3). Results are representative of three independent experiments. (**B**) MTT assay was performed as described in the Materials and Methods section. Results are expressed as means ± SDs and are representative of three independent experiments. * *p* < 0.05, ** *p* < 0.01 (*n* = 3). (**C**,**D**) To examine the protective role of AMPK in MGO-induced apoptosis, HUVECs infected with Ad-LacZ or Ad-CA-AMPKα1 were exposed to 100 µM MGO for 24 h. (**C**) Cells were applied for Western blotting analysis. The graph shows the densitometric quantification of Western blot bands. * *p* < 0.05 vs. vehicle control, # *p* < 0.05 vs. MGO-treated cells (*n* = 3). Results are representative of three independent experiments. (**D**) MTT assay was performed as described in the Materials and Methods section. Results are expressed as means ± SDs and are representative of three independent experiments. * *p* < 0.05, ** *p* < 0.01 (*n* = 3).

**Figure 6 ijms-21-04069-f006:**
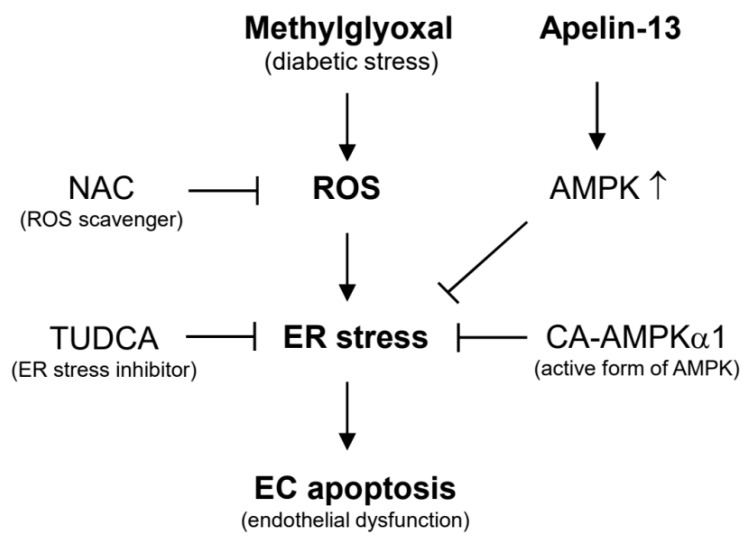
Molecular mechanisms involved in the protective effects of apelin-13 against methylglyoxal-induced endothelial dysfunction. NAC (ROS inhibitor) and TUDCA (ER stress inhibitor) inhibit MGO-induced ER stress and endothelial apoptosis. The activation of AMPK might be involved in the protective effects of apelin-13 against MGO-induced endothelial dysfunction.

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
