# Peer review of "Apelin-13 Inhibits Methylglyoxal-Induced Unfolded Protein Responses and Endothelial Dysfunction via Regulating AMPK Pathway"

_ijms, 2020, doi:10.3390/ijms21114069_

Round 1

Reviewer 1 Report

The auothors have addressed most of my concerns and therefore, I would suggest the current version of the manucript for publication. 

Author Response

We corrected some of typo. 

Thank you very much for your consideration!

Reviewer 2 Report

No further comments

Author Response

We corrected some of typo. 

Thank you very much for your consideration!

This manuscript is a resubmission of an earlier submission. The following is a list of the peer review reports and author responses from that submission.

Round 1

Reviewer 1 Report

The Authors  investigate the mechanism underlying MGO-induced endothelial dysfunction and  examine whether the cytoprotective polypeptide ligand apelin-13, a protects endothelial cells from  MGO damage.

There are a few points that need to be addressed:

1) Dose response experiments need to be performed to establish the most effective dose of both MGO and apelin-13

Response 1: As reviewer suggested, we performed the experiments of dose responses with both MGO and apelin-
13 in the revised manuscript (new Fig 1E, Fig 2A). Based on the results, we used 100 mM MGO and 1 mM apelin-
13 in most of experiments.

2) Western Blot need to be quantified and the results, along with their stastical significancies, need to be presented in bar graph form

Response 2: We quantified densities of Western blot data by densitometry and presented the quantification in bar
graph in the revised manuscript.

3) Overall the quality of the WB images is quite low. This is particularly true for pPERK (fig.1A), pJNK e Jnk tot (fig.1B), ATF-4 and CHOP (fig.2c) and cleaved caspase-3 (fig.5A). 

Response 3: We agree reviewer’s comment and repeated the experiments and replaced higher resolution images
in the revised manuscript (new Figs. 1A, 1B, 2A, 5A).

We hope we have now fully and appropriately addressed all of the reviewer concerns in a satisfactory
manner. However, we would be happy to further address any concerns the reviewer may have.

Reviewer 2 Report

-Line 51: just check typo “can keep a (include this one) homeostatic balanced status.”

Response 1: We agree reviewer’s comment and corrected typo.

-Line 57-58: please rewrite, the wording is difficult to read.

Response 2: We agree reviewer’s comment and rewrite as follow.
However, if the unmanageable demand of protein synthesis is imposed on the ER folding machinery, cell
undergoes to degenerative pathway leading to apoptosis.

-ROS is not mentioned in the introduction, it might be nice to include something about it since there are results relating to this or it could also be a short introduction or rational of the experiment looking at ROS in results part.

 Response 3: We agree reviewer’s comment and describe the role of ROS in the regulation of ER stress in thesection of Introduction.

-The dilutions of antibodies used in the WBs should be indicated.

Response 4: As reviewer recommended, we provided the usage of antibodies for WB in the section of Materials
and Methods.

There is no indication in animal age or if gender was always the same, it would be nice at least to indicate the age since it is an important factor for ER stress

Response 5: As reviewer recommended, we described the animal age and gender in the section of Materials and
Methods as eight-week old male C57BL/6 mice.

- In Fig1A p-PERK seems to disappear at 180 min and reappear at 360min however there is no explanation found anywhere for this. The same applies for p-JNK.

Response 6: We agree reviewer’s comment and repeated the experiments. We did not observe the biphasic
responses of PERK and JNK in most of results, and thus replaced the representative data in the revised manuscript.

- In general WBs are at least n3 experimentally speaking, and a representative WB is added in each of the figures, however I believe a graph with intensities normalized by tubulin should be added in most of the cases since sometimes the difference is not so clear in the image provided.

Response 7: We quantified densities of Western blot data by densitometry and presented the quantification in bar
graph in the revised manuscript.

-Fig1A to 1E have not well indicated the number of experiments as indicated for Fig1F.

Response 8: As reviewer suggested, we indicated the number of experiments in the Figure legends.

-In Fig3A there is no scale bar. It should also be included a quantification of the number of positive cells found in each sample in percentage.

Response 9: As reviewer recommended, we indicated scale bar and bar graph in Fig3A.

Point 10: Line 163 3.1 Authors don’t indicate the cell line they are working at the beginning, it might be
necessary to include it at the beginning of the paragraph.

Response 10: As reviewer suggested, we mentioned the usage of HUVECs in our experiments.

Point 11: Line 189 should be used in plural “apoptotic markers induced by MGO were markedly inhibited”.

Response 11: We are very sorry for inconvenience and corrected the typo.

Point 12: Line 192: I am not sure if is correct to say “protein expression” instead of presence.

Response 12: We agree reviewer’s comment and changed ‘protein expression’ to ‘protein levels’ in the revised
manuscript.
Point 13: Line 208: Typo in “Aortic endothelial cells were (needs to be included) stained with anti-VE
cadherin…”.

Response 13: We are very sorry for inconvenience and corrected the typo.

Point 14: Line 209 “As shown in fig 2D” there is no figure 2D please eliminate this phrase.

Response 14: We are very sorry for inconvenience and eliminate this phrase.

Point 15: Line 232 Please include the meaning of each letter from the acronym NAC.

Response 15: As reviewer suggested, we described the meaning of NAC as N-acetyl cysteine.

Point 16: Fig 4C, ATF4 only seems down after 1h of treatment however it seems to come back at 3h. There is
again no explanation to this phenomenon (also shown in CHOP) and they declare that ATF4 goes down in presence of NAC when in the WB this does not seem to be the case at 3 and 9h. Maybe a graph would help with
this, as I said before, however the change is not that obvious as is the case of CHOP.

Response 16: We agree reviewer’s comment and repeated the experiments and added the bar graph in the revised
manuscript (Fig. 4C).

Point 17: Line 242 and 243 Description of the assay is not very clear, please rephrase these two lines.

Response 17: We agree reviewer’s comment and rephrase the sentence in the Figure 4 legend.

Point 18: Author’s did not present the transduction efficiency.

Response 18: We agree reviewer’s comment and describe the transduction efficiency in the section of Materials
and Methods. Transduction efficiency of adenoviruses were determined by visualization of GFP from Ad-LacZGFP
using the confocal microscope. 10 MOI of Ad-LacZ-GFP showed more than 70 % of green fluorescence
positive cells.

Point 19: The manuscript benefits from a schematic figure summarizing all the interactions analyzed through the
experiments with all components (AMPK, MGO, CHOP, apelin etc).

Response 19: We agree reviewer’s comment and provided the schematic diagram in the Fig 6.

Point 20: The discussion part lacks the in depth discussing the finding in context of previous studies. It does not
go through the experiments deeply but only mentioning figure 4 and 5 without discussing further on. Conclusion
also needs to be written better.

Response 20: We agree reviewer’s comment and described the meaning of findings and added the Discussion and
Conclusion regarding roles of ROS, AMPK and Ca2+ mobilization in MGO-induced endothelial dysfunction.

We hope we have now fully and appropriately addressed all of the reviewer concerns in a satisfactory manner. However, we would be happy to further address any concerns the reviewer may have.